Protein kinase C in the wood frog, Rana sylvatica: reassessing the tissue-specific regulation of PKC isozymes during freezing

Dieni Christopher A. christopher.dieni@carleton.ca
Storey Kenneth B.
Institute of Biochemistry, Carleton University , Ottawa, Ontario , Canada
Druzinsky Robert
Electronic publication date: 2014 Sep 4
Publication date: 2014
Volume: 2
Electronic Location ID: e558
Received 2014 May 19; Accepted 2014 Aug 14
Copyright: © 2014 Dieni and Storey
Copyright year: 2014
Copyright holder: Dieni and Storey
License: This is an open access article distributed under the terms of the Creative Commons Attribution License, which permits unrestricted use, distribution, reproduction and adaptation in any medium and for any purpose provided that it is properly attributed. For attribution, the original author(s), title, publication source (PeerJ) and either DOI or URL of the article must be cited.
License URL: https://creativecommons.org/licenses/by/4.0/

Keywords: Protein kinase C, Phosphorylation, Wood frog, Freeze-tolerance, Tissue-specific, Immunoblotting, Signal transduction, Adaptation, Catalytic competence, Second messenger

Funding: Natural Sciences and Engineering Research Council of Canada OPG 6793 Canada Research Chair in Molecular Physiology Carleton University This work was supported by a discovery grant to KBS from the Natural Sciences and Engineering Research Council of Canada (OPG 6793), and the Canada Research Chairs program (Canada Research Chair in Molecular Physiology). CAD held an Ontario Graduate Scholarship in Science and Technology (OGSST) and a Fluorosense Inc. Scholarship from Carleton University. The funders had no role in study design, data collection and analysis, decision to publish, or preparation of the manuscript.

==============================
The wood frog, Rana sylvatica, survives whole-body freezing and thawing each winter. The extensive adaptations required at the biochemical level are facilitated by alterations to signaling pathways, including the insulin/Akt and AMPK pathways. Past studies investigating changing tissue-specific patterns of the second messenger IP3 in adapted frogs have suggested important roles for protein kinase C (PKC) in response to stress. In addition to their dependence on second messengers, phosphorylation of three PKC sites by upstream kinases (most notably PDK1) is needed for full PKC activation, according to widely-accepted models. The present study uses phospho-specific immunoblotting to investigate phosphorylation states of PKC—as they relate to distinct tissues, PKC isozymes, and phosphorylation sites—in control and frozen frogs. In contrast to past studies where second messengers of PKC increased during the freezing process, phosphorylation of PKC tended to generally decline in most tissues of frozen frogs. All PKC isozymes and specific phosphorylation sites detected by immunoblotting decreased in phosphorylation levels in hind leg skeletal muscle and hearts of frozen frogs. Most PKC isozymes and specific phosphorylation sites detected in livers and kidneys also declined; the only exceptions were the levels of isozymes/phosphorylation sites detected by the phospho-PKCα/βII (Thr638/641) antibody, which remained unchanged from control to frozen frogs. Changes in brains of frozen frogs were unique; no decreases were observed in the phosphorylation levels of any of the PKC isozymes and/or specific phosphorylation sites detected by immunoblotting. Rather, increases were observed for the levels of isozymes/phosphorylation sites detected by the phospho-PKCα/βII (Thr638/641), phospho-PKCδ (Thr505), and phospho-PKCθ (Thr538) antibodies; all other isozymes/phosphorylation sites detected in brain remained unchanged from control to frozen frogs. The results of this study indicate a potential important role for PKC in cerebral protection during wood frog freezing. Our findings also call for a reassessment of the previously-inferred importance of PKC in other tissues, particularly in liver; a more thorough investigation is required to determine whether PKC activity in this physiological situation is indeed dependent on phosphorylation, or whether it deviates from the generally-accepted model and can be “overridden” by exceedingly high levels of second messengers, as has been demonstrated with certain PKC isozymes (e.g., PKCδ).

Introduction

For animals living in boreal climates, cold temperatures, particularly sustained periods of subzero temperatures for months at a time, present a challenge to survival. For many of these animals, the solution is migration or retreating to warmer zones until temperatures in their boreal homes rise once again. For other animals, however, migration of this scope is not possible, and unique arrays of adaptive mechanisms are utilized to endure the prolonged cold. One such animal is the wood frog, Rana sylvatica (reviewed in Storey & Storey, 1996). Each winter, this anuran endures whole-body freezing; approximately 65–70% of extracellular and extra-organ water freezes in the form of nucleated ice, via the actions of ice-nucleating proteins or ice-structuring proteins. During this time, cerebral and cardiovascular activities are undetectable by conventional means. Intracellular freezing and any resulting irreparable damage to cellular contents is prevented by natural cryoprotection; liver glycogen stores undergo extensive hydrolysis (causing a decrease in liver mass by approximately 45%), and glucose is exported and systemically distributed, accumulating in some tissues at levels up to 40–60 times higher than euglycemic levels (Storey & Storey, 1985; Costanzo, Lee & Lortz, 1993). Such a broad reorganization requires numerous modulations at several levels of the signaling and metabolic hierarchy of glucose metabolism, including: (1) phosphorylation and sustained activation of liver glycogen phosphorylase (Crerar, David & Storey, 1988; Mommsen & Storey, 1992); (2) adaptations to plasma membranes in order to facilitate glucose transport and distribution (King, Rosholt & Storey, 1993); (3) tissue-specific adjustment of anabolic and catabolic signaling pathways (e.g., the insulin/Akt pathway, and the adenosine monophosphate-activated protein kinase or AMPK pathway) to optimize glucose production, distribution, uptake, and utilization as a cryoprotectant (Rider et al., 2006; Dieni, Bouffard & Storey, 2012; Zhang & Storey, 2013; do Amaral, Lee & Costanzo, 2013), and; (4) suppression of metabolic pathways that would otherwise divert glucose away from cryoprotection (e.g., pentose phosphate pathway, glycolysis; Dieni & Storey, 2010; Dieni & Storey, 2011), among others. Following the return of warmer temperatures and the arrival of spring, frogs thaw and resume their natural life cycle with no apparent debilitating results of the freeze-thaw process.

Given the scope of these necessary adaptations it is likely, and has in fact already been demonstrated, that altered signaling comprises a major facet of the mechanisms behind the biochemical outcomes facilitating survival. In addition to those signaling enzymes already referenced (i.e., Akt, AMPK, glycogen synthase kinase-3 or GSK3, protein kinase A or PKA), additional kinases and phosphatases have been shown to play a role in wood frog freeze-tolerance. For instance, mitogen activated protein kinases (MAPKs) are activated in various tissues and are suggested as having a role in regulating metabolic or gene expression responses that would facilitate survival in the freezing and/or thawing processes (Greenway & Storey, 2000). Past studies have also suggested a potential role for protein kinase C (PKC) in freezing, anoxia, and dehydration, based on patterns of inositol 1,4,5-trisphosphate (IP3), a second messenger associated with cytosolic calcium increases and a co-product of diacylglycerol (DAG; Holden & Storey, 1996; Holden & Storey, 1997). Increases in cytosolic calcium and DAG both lead to PKC activation.

PKC in fact consists of a family of 15 serine/threonine-protein kinase isozymes in humans, divided into subfamilies with specific second messenger requirements and upstream regulators (Mellor & Parker, 1998); in genome-sequenced amphibians (i.e., Xenopus), the NCBI gene database contains entries for sequences identified as PKCα, PKCβ, PKCγ, PKCδ, PKCε, PKCζ, PKCη, PKCθ, PKCι, PKD/PKCμ. Our lab has previously demonstrated in vivo roles for PKC in various forms of animal stress physiology, including: (1) reptilian anaerobiosis (Mehrani & Storey, 1996); (2) mammalian hibernation (Mehrani & Storey, 1997), and; (3) fish exercise and bioenergetics (Brooks & Storey, 1998). Meanwhile, in vitro stimulation of endogenous PKC has been shown to significantly affect the kinetic properties of glucose-6-phosphate dehydrogenase (G6PDH; Dieni & Storey, 2010), and hexokinase (Dieni & Storey, 2011) from wood frog tissue extracts. Given the potential importance of PKC in wood frog freeze-tolerance, the present study further explores the regulation of this family of kinases in vivo, using phospho-specific immunoblotting to establish tissue-specific phosphorylation states of the PKC isozymes in control and frozen frogs.

Materials and Methods

Animals

Conditions for animal care, experimentation, and euthanasia were approved by the Carleton University Animal Care Committee (B09-22) in accordance with guidelines set down by the Canadian Council on Animal Care. Male wood frogs were captured from spring breeding ponds in the Ottawa, Ontario area. Animals were washed in a tetracycline bath, and placed in plastic containers with damp sphagnum moss at 5 °C for two weeks prior to experimentation. Control frogs were sampled from this condition. For freezing exposure, frogs were placed in closed plastic containers with damp paper toweling on the bottom, and put in an incubator set at −3 °C. A 45 min cooling period was allowed during which body temperature of the frogs cooled to below −0.5 °C and nucleation was triggered due to skin contact with ice crystals formed on the paper toweling (Storey & Storey, 1985). Subsequently, timing of a 24 h freeze exposure began. All frogs were sacrificed by pithing, followed by rapid dissection, and flash-freezing of tissue samples in liquid nitrogen. Tissues were then stored at −80 °C until use.

Tissue extract preparation for SDS-PAGE and immunoblotting

Soluble protein extracts were prepared from tissues that had been previously stored at −80 °C. Briefly, samples of frozen tissues were weighed and then quickly homogenized using a Polytron PT1000 homogenizer (Brinkmann Instruments, Rexdale, ON, Canada) at 50% of full power in a 1:5 w:v ratio with ice-cold buffer A (20 mM Hepes, 200 mM NaCl, 0.1 mM EDTA, 10 mM NaF, 1 mM Na3V O4, and 10 mM ß-glycerophosphate). Protease and phosphatase inhibitors were added just prior to homogenization: 1:1,000 v:v protease inhibitor cocktail (P8340; Sigma, St. Louis, MO, USA), 1:1,000 v:v phosphatase inhibitor cocktail 1 (P2850; Sigma, St. Louis, MO, USA), and a few crystals of phenylmethylsulfonyl fluoride (PMSF). Samples were centrifuged at 10,000 × g for 15 min at 4 °C and then supernatants were removed and held on ice.

Soluble protein concentration was quantified by the Bradford assay (Bradford, 1976) using the Bio-Rad Protein Assay Dye Reagent Concentrate (500-0006; Bio-Rad, Hercules, CA, USA), according to the manufacturer’s instructions, and a Dynatech MR5000 microplate reader (DYNEX Technologies Inc., Chantilly, VA) set at 595 nm. Samples were then adjusted to equal soluble protein concentrations by the addition of small volumes of buffer A; this compensates for differences in the wet:dry ratio of tissues from control versus frozen frogs. Aliquots were mixed 1:1 v:v with SDS-PAGE sample buffer containing: 100 mM Tris-HCl (pH 6.8), 4% w:v sodium dodecyl sulfate (SDS), 20% v:v glycerol, 5% v:v 2-mercaptoethanol and 0.2% w:v bromophenol blue. Following boiling for 5 min, samples were cold-snapped on ice, and stored at −20 °C until use.

SDS-PAGE and polyvinylidene difluoride membrane transfer

Aliquots of thawed samples containing 20 µg of protein were loaded into wells of SDS-polyacrylamide gels (8% resolving gel, 5% stacking gel, made from a 30% acrylamide and bis-acrylamide solution, 37.5:1; 161-0158; Bio-Rad, Hercules, CA, USA), along with Kaleidoscope prestained markers (161-0324; Bio-Rad, Hercules, CA, USA) as a guide for the approximate molecular weight of PKC isozymes. On a typical 12-laned gel, 5 independently-prepared protein extracts from control frogs, and 5 independently-prepared protein extracts from frozen frogs, were loaded in parallel (along with prestained markers); thus, for any given PKC isozyme or specific phosphorylation site being probed, all immunoreactive bands from both control and experimental animals detected at the chemiluminescence stage will have been treated identically through all electrophoretic, transfer, immunoblotting, and chemiluminescence/exposure steps. Samples were electrophoresed at 180 V in a Mini-PROTEAN III apparatus (Bio-Rad, Hercules, CA, USA) using 1x running buffer (5x running buffer contained 15.1 g Tris-base, 94 g glycine, and 5 g SDS per litre, pH 8.3). Proteins were then wet-transferred to polyvinylidene difluoride (PVDF) membrane (Millipore, Bedford, MA, USA) using a current of 300 mA for 1.5 h at 4 °C in a Bio-Rad Mini Trans-Blot Cell apparatus (Bio-Rad, Hercules, CA, USA). Transfer buffer contained 25 mM Tris-base pH 8.8, 192 mM glycine, and 20% v:v methanol, chilled to 4 °C.

Immunoblotting of PVDF membranes and analysis

Primary antibodies (Cell Signalling Technology, Danvers, MA, USA) were the following: phospho-PKC (pan) (βII Ser660) antibody (9371), which detects all of PKCα, βI, βII, δ, ε and η isoforms only when phosphorylated at a carboxy-terminal residue homologous to Ser660 of PKCβII; phospho-PKCδ/θ (Ser643/676) antibody (9376), which detects both PKCδ when phosphorylated at Ser643 and PKCθ when phosphorylated at Ser676; phospho-PKCα/βII (Thr638/641) antibody (9375); phospho-PKCδ (Thr505) antibody (9374; this antibody has since become unavailable after this work was carried out); phospho-PKCθ (Thr538) antibody (9377); phospho-PKCζ/λ (Thr410/403) antibody (9378); PKD/PKCμ antibody (2052); phospho-PKD/PKCμ (Ser916) antibody (2051); phospho-PKD/PKCμ (Ser744/748) antibody (2054). All primary IgG antibodies were raised in rabbit. These were purchased together as the Phospho-PKC Antibody Sampler Kit (9921; this kit has since become unavailable after this work was carried out). Stock primary antibodies were diluted between 1:5,000 and 1:10,000 in Tris-buffered saline supplemented with Tween-20 (TBST; 20 mM Tris pH 7.5, 150 mM NaCl, 0.05% v:v Tween-20). Secondary antibody used was the anti-rabbit IgG, HRP-linked antibody (7074; also supplied within the Phospho-PKC Antibody Sampler Kit). Stock secondary antibodies were diluted 1:2,000 in TBST. We opted to use these antibodies, focusing on phospho-PKC and not unphosphorylated forms of PKC, for two main reasons. Firstly, given that these antibodies were specifically distributed as an assembled kit (at that time), we were hesitant to introduce additional antibodies that had possibly been developed, raised, and purified differently (potentially even from different commercial sources) from those provided in the kit. Secondly, as will be further detailed in the Discussion section, the scope of this study followed the widely-accepted model that only phosphorylated PKC is catalytically active; we therefore were especially interested in phospho-specific forms, so as to relate our previous forays into PKC second messengers (Holden & Storey, 1996; Holden & Storey, 1997) to resulting effects on PKC phosphorylation states in frozen frogs.

After transfer was complete, PVDF membranes were typically cut using a razor blade, so as to allow parallel immunoblotting of several different frog proteins (these were unrelated to the current study) using multiple antibodies but with a single starting tissue extract and PVDF membrane (Silva & McMahon, 2014). This practice permits efficient utilization of tissue and protein extract resources, particularly when the model organism under study is small and tissues are limiting; the male wood frog typically has a body mass of 4–7 g, and in dehydration studies (one example of the Rana sylvatica stress–tolerance studies conducted by our group) frogs will only be sacrificed and dissected once they have lost ∼40% of their total body water (Abboud & Storey, 2013). These PVDF membrane sections were quickly equilibrated in TBST and then blocked with 5% w:v nonfat milk dissolved in TBST for 15 min at room temperature. The blot was rinsed with TBST and then incubated with primary antibody in TBST on a shaking platform overnight at 4 °C. Blots were washed twice with TBST and incubated with secondary antibody for 1.5 h at room temperature. Immunoreactive bands, on blots consisting of protein transfers from 5 independently-prepared protein extracts from control frogs and 5 independently-prepared extracts from frozen frogs, were visualized using enhanced chemiluminescence (ECL; RPN2108, GE Healthcare Life Sciences, Baie d’Urfé, QC, Canada) following the manufacturer’s protocol. The luminol and oxidizing reagents were mixed 1:1 v:v on the membrane for 1 min and the ECL signal was detected using a ChemiGenius (SynGene, Frederick, MD, USA).

Total protein was then visualized on the PVDF membrane by staining for 30 min with Coomassie blue staining solution (0.25% w:v Coomassie Brilliant Blue R, 50% v:v methanol, 7.5% v:v acetic acid) followed by destaining with destain solution (25% v:v methanol, 10% v:v acetic acid). Three Coomassie-stained bands that did not differ in intensity between active and frozen conditions were used to normalize the corresponding intensity of the immunoreactive band in each lane to correct for any unequal protein loading, as described previously (Dieni, Bouffard & Storey, 2012). Our group typically opts to follow this method of protein normalization, instead of probing for “conventional” loading controls such as actin or tubulin, for all our stress-physiology and adaptation studies (Abboud & Storey, 2013; Lama, Bell & Storey, 2013; Rouble et al., 2013); this is an increasingly-common practice in other groups (Goldberg et al., 2013; Bahar et al., 2014; Da’dara et al., 2014), particularly in instances where levels of housekeeping proteins themselves are suspected of changing due to pharmacological, pathophysiological, or physiological stress (Li et al., 2011; Eaton et al., 2013; Parrondo et al., 2013).

Intensities of ECL-visualized and Coomassie-stained bands were quantified using the associated Gene Tools program (v. 3.00.02). Data were analyzed by one-way ANOVA followed by Tukey’s test; a statistically-significant difference was accepted with values of p < 0.05 or smaller.

Results and Discussion

Overall scope of phospho-PKC levels and changes in freezing

The widely-accepted model for activation of PKC isozymes has been reviewed (Mellor & Parker, 1998; Parker & Murray-Rust, 2004; Gomperts, Kramer & Tatham, 2009) and follows here. PKCs are biosynthesized as catalytically inactive, and must first bind to the intracellular face of the plasma membrane in order to be unfolded, and rendered competent. A number of upstream signals can activate phospholipases, hydrolysing inositol phospholipids to various combinations of diacylglycerols and IP3; IP3 will in turn trigger calcium efflux from the endoplasmic reticulum, which then propagates further calcium influx from the extracellular environment. Conventional PKC isozymes (cPKC; α, β, γ) bind to the membrane via two specific bridging interactions: C1 domains that bind to DAG and C2 domains that bind to calcium-phospholipid complexes. Once bound, cPKCs unfold such that their hydrophobic motifs interact with and activate 3-phosphoinositide dependent protein kinase-1 (PDK1), and the PKC pseudosubstrate motif is withdrawn from its catalytic core. PDK1, currently the single conclusive upstream kinase responsible for phosphorylation of the PKC activation loop (Le Good et al., 1998; Ron & Kazanietz, 1999), phosphorylates this loop and triggers two successive autophosphorylations, one on the PKC turn motif, and one on the hydrophobic motif. Only once at this stage, phosphorylated at three sites and bound to both DAG and calcium, have cPKCs typically been recognized as fully-active. Depletion of DAG and calcium will induce cPKC refolding and inactivation; however, as long as the aforementioned sites remain phosphorylated, cPKCs can be instantly reactivated upon reintroduction of DAG and calcium. Thus, for cPKCs, all three criteria of DAG, calcium, and upstream phosphorylation are often necessary for full activity; this is complicated for novel PKC isozymes (nPKC; δ, ε, η, θ) and atypical PKC isozymes (aPKC; ζ, λ, μ). nPKCs are calcium-independent, but still rely on DAG for activity; aPKCs rely on neither calcium nor DAG, or any other phospholipids, though they possess other unique domains such as the phox-bem1 domain, which suggests that protein–protein interactions with cytosolic partners may be necessary for activity (Gomperts, Kramer & Tatham, 2009). In both these PKC subfamilies however, phosphorylation is typically a prerequisite for catalytic competence. It should be noted, however, that while there is nearly three decades’ worth of literature investigating phosphorylation as a prerequisite for what we refer to here as “catalytic activity” (Parekh, Ziegler & Parker, 2000; Wang et al., 2012; Parker et al., 2014), this model is coming under increasing scrutiny (Wu-Zhang & Newton, 2013), and some studies have even pointed to PKC phosphorylation as leading to degradation rather than activation (Brand et al., 2010). The scope of our discussion will focus primarily on the widely-accepted model presented earlier, whereby PKC phosphorylation leads to “activation” whereby an increase in catalytic activity has typically been observed.

Extracts of hind leg skeletal muscle, liver, heart, kidney, and brain from control and frozen frogs were probed with all 9 primary antibodies of the Phospho-PKC Antibody Sampler kit; however, not all antibodies revealed the presence of immunoreactive bands in each tissue extract (e.g., only 2 out of the 9 primary antibodies revealed bands in muscle homogenates). In each case where antibodies detected bands, only a single and distinct band appeared in the area of our cut PVDF membrane section; immunoreactive bands were confirmed to be PKC isozymes by comparing their approximate molecular weights to those listed on the manufacturer’s datasheet provided (http://www.cellsignal.com/pdf/9921.pdf). A summary of changes between control and frozen frogs is presented in Table 1. In the case of each individual antibody, 5 immunoreactive bands from independently-prepared control frog protein extracts, and 5 from independently-prepared frozen frog protein extracts, were quantified from the same immunoblot under the same exposure conditions, and for the purposes of clarity 2 bands from each physiological state (i.e., control vs. frozen) were presented in Figs. 1–5. In general, levels of phosphorylated PKC isozymes (and non-phosphorylated PKD/PKCμ) tended to globally decrease during wood frog freezing in hind leg skeletal muscle, liver, kidney, and heart; the only tissue in which increases in phospho-PKC were observed was the brain.

Figure 1 Changes in phosphorylation levels of PKC isozymes in frog hind leg skeletal muscle during freezing.

(A) Relative levels were determined from immunoblots of n = 5 independently-prepared tissue homogenates from pooled tissues of either control frogs, or frogs frozen for 24 h. 2 representative bands out of the 5 total bands for both control and frozen frogs are included in this figure. (B) Densitometry of immunoreactive bands as quantified by the Gene Tools program. Closed (black) bars represent data from control frogs, whereas open (white) bars represent data from frozen frogs. Statistically significant differences, determined by one-way ANOVA followed by Tukey’s test, are as follows: *, p < 0.005; **, p < 0.001.

Figure 2 Changes in phosphorylation levels of PKC isozymes in frog liver during freezing.

(A) Relative levels were determined from immunoblots of n = 5 independently-prepared tissue homogenates from pooled tissues of either control frogs, or frogs frozen for 24 h. 2 representative bands out of the 5 total bands for both control and frozen frogs are included in this figure. (B) Densitometry of immunoreactive bands as quantified by the Gene Tools program. Closed (black) bars represent data from control frogs, whereas open (white) bars represent data from frozen frogs. Statistically significant differences, determined by one-way ANOVA followed by Tukey’s test, are as follows: *, p < 0.05; **, p < 0.005; ***, p < 0.001. † represents quantifications where immunoreactive bands were not detectable in liver extracts of frozen frogs.

Figure 3 Changes in phosphorylation levels of PKC isozymes in frog kidney during freezing.

(A) Relative levels were determined from immunoblots of n = 5 independently-prepared tissue homogenates from pooled tissues of either control frogs, or frogs frozen for 24 h. 2 representative bands out of the 5 total bands for both control and frozen frogs are included in this figure. (B) Densitometry of immunoreactive bands as quantified by the Gene Tools program. Closed (black) bars represent data from control frogs, whereas open (white) bars represent data from frozen frogs. Statistically significant differences, determined by one-way ANOVA followed by Tukey’s test, are as follows: *, p < 0.05; **, p < 0.005; ***, p < 0.001.

Figure 4 Changes in phosphorylation levels of PKC isozymes in frog heart during freezing.

(A) Relative levels were determined from immunoblots of n = 5 independently-prepared tissue homogenates from pooled tissues of either control frogs, or frogs frozen for 24 h. 2 representative bands out of the 5 total bands for both control and frozen frogs are included in this figure. (B) Densitometry of immunoreactive bands as quantified by the Gene Tools program. Closed (black) bars represent data from control frogs, whereas open (white) bars represent data from frozen frogs. Statistically significant differences, determined by one-way ANOVA followed by Tukey’s test, are as follows: *, p < 0.05; **, p < 0.01; ***, p < 0.005. † represents quantifications where immunoreactive bands were not detectable in heart extracts of frozen frogs.

Figure 5 Changes in phosphorylation levels of PKC isozymes in frog brain during freezing.

(A) Relative levels were determined from immunoblots of n = 5 independently-prepared tissue homogenates from pooled tissues of either control frogs, or frogs frozen for 24 h. 2 representative bands out of the 5 total bands for both control and frozen frogs are included in this figure. (B) Densitometry of immunoreactive bands as quantified by the Gene Tools program. Closed (black) bars represent data from control frogs, whereas open (white) bars represent data from frozen frogs. Statistically significant differences, determined by one-way ANOVA followed by Tukey’s test, are as follows: *, p < 0.01. † represents quantifications where immunoreactive bands were not detectable in brain extracts of control frogs.

Table 1 Summary of changes in phosphorylation levels of PKC isozymes (or non-phosphorylated PKD/PKCμ) during wood frog freezing.

Relative levels were determined from immunoblots of n = 5 independently-prepared tissue homogenates from pooled tissues of control or frozen frogs. Quantifiable decreases (−) or increases (+) are presented numerically. An equal sign (=) indicates no significant change. In some instances, bands were detectable in control frogs but were undetectable or too faint to be accurately quantified in frozen frogs (−−), or vice-versa (+ +). ND indicates that bands for that isozyme or phosphorylation site were not detected in that tissue.

	Muscle	Liver	Kidney	Heart	Brain	
Phospho-PKCα/βII (Thr638/641)	−41.8%***	=	=	−28.6%*	+ 121.3%**	
Phospho-PKCδ (Thr505)	ND	−−	−76.6%****	ND	+ +	
Phospho-PKCδ/θ (Ser643/676)	ND	−54.8%*	−75.7%***	−−	=	
Phospho-PKD/PKCμ (Ser744/748)	ND	−−	ND	ND	=	
Phospho-PKD/PKCμ (Ser916)	ND	−−	ND	ND	ND	
PKD/PKCμ	ND	−76.4%****	−66.7%****	−−	=	
Phospho-PKC (pan) (βII Ser660)	ND	−82.7%***	−74.6%*	−38.6%***	=	
Phospho-PKCθ (Thr538)	−50.4%****	−−	ND	−35.8%**	+ +	
Phospho-PKCζ/λ (Thr410/403)	ND	ND	−66.8%****	−−	ND	
Notes.

Statistical significance as determined by one-way ANOVA followed by Tukey’s test is as follows.

* p < 0.05.

** p < 0.01.

*** p < 0.005.

**** p < 0.001.

PDK1 itself, and its targets, have been shown to change in phosphorylation state during wood frog freezing. For instance, levels of phospho-Thr308-Akt (a phosphorylation site of PDK1) decrease in muscle and heart during freezing, suggesting decreased action of PDK1 in these tissues. By contrast, levels of both phospho-Ser241-PDK1 and phospho-Thr308-Akt increase in liver, suggesting increased PDK1 action in livers of frozen frogs (Zhang & Storey, 2013). For optimal clarity, specific changes in PKC isozymes will be further described and discussed on a tissue-by-tissue basis, and compared to previously-established changes in PDK1 or its targets, or previously-assessed targets downstream of PKC isozymes.

Muscle

Only 2 primary antibodies were immunoreactive to frog muscle extracts: phospho-PKCα/βII (Thr638/641), and phospho-PKCθ (Thr538). Bands detected by both these antibodies decreased in intensity in frozen frogs (Fig. 1; Table 1). Thr638/641 is in the turn motif of PKCα/βII, and is autophosphorylated after initial phosphorylation of the activation loop by PDK1 (Ron & Kazanietz, 1999). Whether phospho-Thr638/641 is necessary for PKCα/βII activity is debatable; rather, it is more recognized for its importance in duration of PKC activation, and slowing the rate of PKC activation loop dephosphorylation (Bornancin & Parker, 1996). Meanwhile, Thr538 is in the activation loop of PKCθ, and is directly phosphorylated by PDK1; as such, it is unequivocally needed for PKCθ activity (Liu et al., 2002). Taken together, these results suggest a combination of lower activity, a shorter duration of activation, and a higher rate of dephosphorylation of these PKC isozymes in muscle during freezing.

These decreases in the phosphorylation levels of PKC muscle isozymes correlate well with recently-presented decreases in phospho-Thr308-Akt (a target that, along with PKC, is also a direct substrate of PDK1; Zhang & Storey, 2013). By contrast, past studies would suggest a need for PKC to remain active in muscle; these have shown that in response to freezing, IP3 levels rose moderately, by 55%, in skeletal muscle (Holden & Storey, 1996; Holden & Storey, 1997). Moreover, calcium binding and uptake into the sarcoplasmic reticulum were strongly decreased in skeletal muscle of frozen frogs, leading to increased cytosolic calcium levels (Hemmings & Storey, 2001). However, in yet other studies, muscle IP3 levels remained constant in frogs subjected to short-term anoxia at 5 °C, and then fell by 40% after 2 days of anoxic exposure. The contrasting past and present findings of increased IP3 and calcium levels in frozen frogs, yet decreased PKCα/βII and PKCθ phosphorylation in this same physiological state, along with decreased IP3 levels in frogs subjected to long-term anoxia, leaves us with a very uncertain role for muscle PKC in the adaptation to these stresses.

Liver

8 of the 9 primary antibodies were immunoreactive to frog liver extracts; only the phospho-PKCζ/λ (Thr410/403) antibody failed to reveal any bands. Overall, band intensities again tended to decrease in frozen frogs (Fig. 2; Table 1). As in muscle, phospho-Thr538-PKCθ levels decreased, but to such an extent where they were non-quantifiable in frozen frogs. Similarly, phospho-Thr505-PKCδ levels also decreased to an extent where they were non-quantifiable. Interestingly, in contrast to Thr538, an activation loop phosphorylation of PKCθ which is unequivocally needed for activity, Thr505 is also an activation loop residue of PKCδ, but one which is at best only debatably necessary for activity, and is autophosphorylated in addition to being phosphorylated by PDK1 (Le Good et al., 1998; Liu et al., 2002; Steinberg, 2004; Liu et al., 2006). Furthermore, phospho-Thr676/643-PKCδ/θ levels (a turn motif phosphorylation) decreased by over 50%. While the role of this turn motif phosphorylation is inconclusive in PKCδ or PKCθ (Li et al., 1997; Liu et al., 2002), decreases in turn motif phospho-Thr676/643 will potentially compound the decreases in activation loop phospho-Thr505/538, further depressing PKCδ and PKCθ activities.

Additional decreases are observed in non-phosphorylated levels of PKD/PKCμ, of phospho-Ser744/748-PKD/PKCμ, and of phospho-Ser916-PKD/PKCμ. Ser916 is an autophosphorylation site that correlates with catalytic activity in PKD/PKCμ (Matthews, Rozengurt & Cantrell, 1999), whereas Ser744 and possibly Ser748 are activation loop phosphorylation sites, also critical to activity (Waldron et al., 2001; Waldron & Rozengurt, 2003). Interestingly, while PKD/PKCμ was originally classified as a member of the PKC family (and is still very much considered as such), Ser744/748 is in fact phosphorylated by other PKC isozymes upstream of PKD/PKCμ, most notably PKCδ (Waldron et al., 2001; Waldron & Rozengurt, 2003). Given that total and phospho-levels of PKD/PKCμ and upstream PKCδ all decreased, these suggest that PKD/PKCμ will also be inactive in frozen frogs.

Wood frog liver is quite possibly the best-characterized tissue in terms of proteins and genes that may have potential relationships with PKC; by virtue of their decreased phosphorylation, the potential decline of PKCδ, PKCθ, and PKD/PKCμ activities contrast with previous findings. One early study demonstrated a progressive rise in IP3 levels over the course of the freezing process, ultimately rising 11-fold higher than control values after 24 h of freezing (Holden & Storey, 1996). Based on this finding, important roles were suggested for PKC, including: (i) overriding of normal cellular metabolic controls; (ii) enabling “acceptance” of prolonged and extreme reductions in cell volume, along with accompanying hyperosmolality and elevated ionic strength, and; (iii) maintaining glycogen phosphorylase in a highly-active state, driving glycogenolysis forward for the degree of hyperglycemia seen in frozen frogs. A follow-up study, exploring second messenger changes in frogs subjected to dehydration and anoxia, noted increased IP3 levels in both dehydrated and anoxic frogs (Holden & Storey, 1997). Because of the response to both dehydration and anoxia, the importance of PKC in freezing was reaffirmed; indeed, both dehydration and anoxia result from the freezing of extracellular water in wood frogs.

Upon the discovery of fr47, a novel gene associated with freezing survival, it was noted that its expression pattern paralleled that of IP3 accumulation, suggesting that PKC may activate freeze-response genes (McNally, Sturgeon & Storey, 2003). Later, NFκB, a transcription factor crucial in cellular stress response and survival, was found to have increased DNA-binding affinity in frozen frogs; its sequestering binding partner, IκB, was also found to increase in phosphorylation during freezing (Storey, 2008). IκB is a substrate of the IκB kinase (IKK), which is in turn a substrate of PKC isozymes (Lallena et al., 1999; Diaz-Meco & Mostat, 2012). Recently, it was shown that Nrf2, a transcription factor activated during oxidative stress, has increased DNA-binding affinity in frozen frogs; moreover, transcription of gsta, a gene under Nrf2 control, was elevated in frozen frogs (Zhang, 2013). It was reiterated in this study that PKC phosphorylates Keap1, a sequestering binding partner of Nrf2, inducing dissociation and activation of Nrf2 and the transcription of antioxidant response genes. Lastly, and possibly most conflicting with the presently-observed decrease in liver phospho-PKC levels, are the rise of both PDK1 and Akt phosphorylation levels in frozen frogs (Zhang & Storey, 2013). Given these collective findings, the presently-suggested decline in liver PKC activities is at odds with the previously-inferred importance of PKC in freezing survival mechanisms.

A possible reconciliation is that although decreases are observed in the phosphorylation states of PKCδ, PKCμ, and PKCθ, levels detected by the phospho-Thr638/641-PKCα /βII antibody remain apparently unchanged in frozen frogs. Thr638, a turn motif phosphorylation site, is not critical to PKCα catalytic function but rather controls the duration of its activation by regulating the rate of dephosphorylation and inactivation (Bornancin & Parker, 1996; Li et al., 1997; Ron & Kazanietz, 1999). By contrast, Thr641 is also a turn motif phosphorylation site but is fundamental to the activity of PKCβI and PKCβII (Zhang et al., 1993; Ron & Kazanietz, 1999; Leonard et al., 2011). Taken together, these suggest that some cPKCs will continue to be active and/or remain active for longer in frozen frogs. It should be noted however, that while levels of phospho-Thr638/641-PKCα/βII remained unchanged, levels detected by the phospho-PKC (pan) (βII Ser660) antibody decreased in frozen frogs. This antibody is specific for PKCα, βI, βII, δ, ε and η when autophosphorylated at a carboxy-terminal residue homologous to Ser660 in the hydrophobic motif of PKCβII, following initial phosphorylation of Thr500 by PDK1; phospho-Ser660 plays important roles in correct folding, as well as the binding of protein substrate, ATP, and calcium (Zhang et al., 1993; Ron & Kazanietz, 1999; Leonard et al., 2011). Another possible reconciliation is that while much of the literature supports an activation model where phosphorylation is necessary for PKC activity, there are exceptions; as indicated earlier for instance, the active loop phosphorylation at Thr505-PKCδ is not required for activity (Steinberg, 2004; Liu et al., 2006). As an nPKC, PKCδ, and others that behave similarly, may therefore be active in presence of elevated DAG regardless of phosphorylation state.

Kidney

6 antibodies revealed bands in frog kidney extracts. As in liver, levels of phospho-Thr638/641-PKCα/βII remain unchanged between control and frozen frogs (Fig. 3; Table 1). However, significant decreases were observed in the levels bands detected by phospho-Thr505-PKCδ, phospho-Thr676/643-PKCδ/θ, non-phosphorylated PKD/PKCμ, and phospho-PKC (pan) (βII Ser660) antibodies. Moreover, decreases were also observed in the intensities of the bands detected by the phospho-PKCζ/λ (Thr410/403) antibody. Thr410 is an activation loop residue in PKCζ, as is Thr403 in PKCλ (also known as PKCι in mammals), and these are directly phosphorylated by PDK1 and are critical to activity (Le Good et al., 1998; Le Good & Brindley, 2004). Together, these results would suggest a decreased overall activity for PKC isozymes in kidney of frozen frogs. As with muscle and liver, however, kidney IP3 levels were shown to increase in previous studies (Holden & Storey, 1997). In frogs dehydrated by 40%, IP3 levels rose by 60%; levels remained unchanged in frogs exposed to anoxia (Holden & Storey, 1997) and in frozen frogs (Holden & Storey, 1996). Again, this presents us with potentially contrasting results between second messengers, specific stresses, and PKC phosphorylation state; second messengers responsible for PKC activation increase in kidney (but only in response to dehydration, and not to anoxia or freezing), while PKC phosphorylation itself does not increase during freezing.

Heart

6 antibodies revealed bands in frog kidney extracts, and all of these decreased in frozen frogs (Fig. 4; Table 1). Levels of bands detected by phospho-Thr638/641-PKCα/βII, phospho-Thr676/643-PKCδ/θ, non-phosphorylated PKD/PKCμ, phospho-PKC (pan) (βII Ser660), phospho-Thr538-PKCθ, and phospho-Thr410/403-PKCζ/λ antibodies all decreased significantly; indeed, phospho-Thr676/643-PKCδ/θ, non-phosphorylated PKD/PKCμ, and phospho-Thr410/403-PKCζ/λ levels all decreased to an extent where they were no longer detectable in frozen frogs.

With decreases observed in the phosphorylation levels of all PKC isozymes detected in heart, this would suggest overall decreased PKC activity in this tissue. The decreases in phosphorylation levels of these PKC isozymes also correlate with decreases in phospho-Thr308-Akt in frog heart during freezing (Zhang & Storey, 2013). Previous studies showed no significant changes in IP3 levels in response to freezing (Holden & Storey, 1996), yet significant increases were observed after only 1 h of enduring anoxia (Holden & Storey, 1997). Thus, our present results in heart are in agreement with decreased phosphorylation of other PDK1 targets and unchanged second messenger levels in freezing, but not with increased second messenger levels in response to anoxia.

Brain

8 antibodies revealed bands in frog brain extracts. Of the five tissues investigated in the present study, the results obtained in the brain were the most unique; changes (or lack thereof) in phospho-PKC levels in brain during frog freezing contrasted strongly with those observed in muscle, liver, heart, and kidney (Fig. 5; Table 1). No changes were observed in the levels of bands detected by phospho-Thr676/643-PKCδ/θ, phospho-Ser744/748-PKD/PKCμ, non-phosphorylated PKD/PKCμ, and phospho-PKC (pan) (βII Ser660) antibodies; each of these were detectable in both control and frozen frogs to approximately the same extent. Interestingly, phospho-Thr638/641-PKCα/βII levels increased by 121.3% in brains of frozen frogs. Moreover, phospho-Thr505-PKCδ and phospho-Thr538-PKCθ, which were not detectable in control frogs, were detectable (albeit faintly) in frozen frogs. Overall, whereas phospho-PKC levels generally tended to decline in other organs of frozen frogs, phospho-PKC levels largely remained unchanged or even increased in brain.

The increase in brain PKC phosphorylation during freezing is of great interest. Previous studies have shown that brain IP3 levels rose significantly after 4 h of freezing (Holden & Storey, 1996), and so this is one frog tissue in which the overall increases in PKC phosphorylation state correlate with this rise in IP3. Numerous other adaptive “activations” occur in frog brains in response to freezing, including: (1) moderately-increased expression of fr10 (Cai & Storey, 1997) and li16 (Sullivan & Storey, 2012), novel genes with putative roles in freezing protection; (2) increased levels of c-Fos (Greenway & Storey, 2000), and; (3) up-regulation of genes for ribosomal proteins, including the acidic ribosomal phosphoprotein P0 (Wu & Storey, 2005) and the ribosomal large subunit protein 7 (Wu, De Croos & Storey, 2008). At present, however, we cannot conclusively identify any of the proteins listed here as being substrates of PKC, nor can we confirm that any of the upregulated-genes are facilitated by transcription factors downstream of PKC.

Conceivably, the phosphorylation and activation of PKC (along with other proteins) in brain, contrasted against the decreased or unchanged phosphorylation of PKC in other tissues, suggests a unique and vital role for brain PKC during freezing. Indeed, the importance of PKC in cerebral protection has been demonstrated well-beyond the niche of wood frog freezing (Sun et al., 2013; Thompson et al., 2013). Future efforts will be required to establish the catalytic competence of PKC isozymes in frozen frog brains, to identify the targets being phosphorylated by PKC, and to determine their most likely role in frog cerebral/neuroprotection during freezing based on our current understanding of other models.

Conclusion

The present study investigated the phosphorylation state of conventional, novel, and atypical PKC isozymes in five tissues of freeze-tolerant frogs, as well as non-phosphorylated PKCμ/PKD. Broadly, phospho-PKC levels and non-phosphorylated PKCμ/PKD decreased in muscle, livers, kidneys, and hearts of frozen frogs; the only exception was protein detected by the phospho-Thr638/641-PKCα/βII (turn motif), which showed no change in livers or kidneys between the control and frozen states. This isozyme alone would partially support the findings of past studies, where IP3 was shown to dramatically increase in livers of frozen frogs and thus an important role was suggested for PKC in the freezing process; however, even the steady phosphorylation state of Thr638/641-PKCα/βII is seemingly contradicted by the decreased phosphorylation of Ser660-PKC (pan) (hydrophobic motif) in both livers and kidneys of frozen frogs. A particularly interesting finding in this study is PKC would seem to play an important role in the brains of frozen frogs, as the phospho-levels of all isozymes detected remain either remain unchanged or even increase in the frozen state.

The results of this study succeed in answering some questions of past studies pertaining to the state of PKC in freeze-tolerance, but raise many others and indeed pose some contradictions. Whereas our lab has previously asserted that based on second messenger patterns, PKC would play an important role (particularly in liver and muscle), our present findings would seem to instead suggest a diminished role for PKC in most tissues, based on our current understanding of PKC isozymes and their generally-accepted activation model. This in itself, however, can be debated by pointing to studies which demonstrate that activation loop phosphorylation is, in fact, not needed for PKC activity (e.g., PKCδ). To clarify the role of PKC in wood frog freezing, it is now evident that catalytic assays are necessary in order to unequivocally establish the actual activities of each of these isozymes in the control and frozen states. Moreover, there is an additional difficulty in contextualizing the role of PKC due to the fact that very few downstream targets of PKC have been assessed in wood frogs. Those that were discussed in this report are either only speculative (e.g., transcription factors that might control expression of li16, fr10 and fr47), or are several degrees removed from being direct PKC substrates and/or are under the control of multiple kinases (e.g., NFκB/IκB via IKK, Nrf2 for which Keap1 can be modified via several pathways, etc.) There is a clear need to assess the phosphorylation state of direct PKC substrates (e.g., the MARCKS family) in order to better determine the activity and role of PKC in freeze-tolerance.

Supplemental Information

Supplemental Information 1 Typical immunoblotting pattern for the phospho-PKC (pan) (βII Ser660) antibody

A full blot is presented here for liver, but in other instances PVDF membranes were typically cut at the approximate molecular weight of PKC so that membranes could be used for multiple antibodies (as described in Materials and Methods). 5 independently-prepared protein extracts from control frogs (left) and 5 independently-prepared protein extracts from frozen frogs (right) were electrophoresed, transferred, immunoblotted, and exposed in parallel.

Click here for additional data file.

Supplemental Information 2 Typical immunoblotting pattern for the phospho-PKCα/βII (Thr638/641) antibody

A full blot is presented here for kidney, but in other instances PVDF membranes were typically cut at the approximate molecular weight of PKC so that membranes could be used for multiple antibodies (as described in Materials and Methods). 5 independently-prepared protein extracts from control frogs (left) and 5 independently-prepared protein extracts from frozen frogs (right) were electrophoresed, transferred, immunoblotted, and exposed in parallel.

Click here for additional data file.

Supplemental Information 3 Typical immunoblotting pattern for the phospho-PKCδ (Thr505) antibody

A full blot is presented here for liver, but in other instances PVDF membranes were typically cut at the approximate molecular weight of PKC so that membranes could be used for multiple antibodies (as described in Materials and Methods). 5 independently-prepared protein extracts from control frogs (left) and 5 independently-prepared protein extracts from frozen frogs (right) were electrophoresed, transferred, immunoblotted, and exposed in parallel.

Click here for additional data file.

Supplemental Information 4 Typical immunoblotting pattern for the phospho-PKCδ/θ (Ser643/676) antibody

A full blot is presented here for liver, but in other instances PVDF membranes were typically cut at the approximate molecular weight of PKC so that membranes could be used for multiple antibodies (as described in Materials and Methods). 5 independently-prepared protein extracts from control frogs (left) and 5 independently-prepared protein extracts from frozen frogs (right) were electrophoresed, transferred, immunoblotted, and exposed in parallel.

Click here for additional data file.

Supplemental Information 5 Typical immunoblotting pattern for the phospho-PKCθ (Thr538) antibody

A full blot is presented here for liver, but in other instances PVDF membranes were typically cut at the approximate molecular weight of PKC so that membranes could be used for multiple antibodies (as described in Materials and Methods). 5 independently-prepared protein extracts from control frogs (left) and 5 independently-prepared protein extracts from frozen frogs (right) were electrophoresed, transferred, immunoblotted, and exposed in parallel.

Click here for additional data file.

Supplemental Information 6 Typical immunoblotting pattern for the phospho-PKCζ/λ (Thr410/403) antibody

A full blot is presented here for kidney, but in other instances PVDF membranes were typically cut at the approximate molecular weight of PKC so that membranes could be used for multiple antibodies (as described in Materials and Methods). 5 independently-prepared protein extracts from control frogs (left) and 5 independently-prepared protein extracts from frozen frogs (right) were electrophoresed, transferred, immunoblotted, and exposed in parallel.

Click here for additional data file.

Supplemental Information 7 Typical immunoblotting pattern for the phospho-PKD/PKCμ (Ser744/748) antibody

A full blot is presented here for liver, but in other instances PVDF membranes were typically cut at the approximate molecular weight of PKC so that membranes could be used for multiple antibodies (as described in Materials and Methods). 5 independently-prepared protein extracts from control frogs (left) and 5 independently-prepared protein extracts from frozen frogs (right) were electrophoresed, transferred, immunoblotted, and exposed in parallel.

Click here for additional data file.

Supplemental Information 8 Typical immunoblotting pattern for the phospho-PKD/PKCμ (Ser916) antibody

A full blot is presented here for liver, but in other instances PVDF membranes were typically cut at the approximate molecular weight of PKC so that membranes could be used for multiple antibodies (as described in Materials and Methods). 5 independently-prepared protein extracts from control frogs (left) and 5 independently-prepared protein extracts from frozen frogs (right) were electrophoresed, transferred, immunoblotted, and exposed in parallel.

Click here for additional data file.

Supplemental Information 9 Typical immunoblotting pattern for the PKD/PKCμ antibody

A full blot is presented here for liver, but in other instances PVDF membranes were typically cut at the approximate molecular weight of PKC so that membranes could be used for multiple antibodies (as described in Materials and Methods). 5 independently-prepared protein extracts from control frogs (left) and 5 independently-prepared protein extracts from frozen frogs (right) were electrophoresed, transferred, immunoblotted, and exposed in parallel.

Click here for additional data file.

The authors wish to thank Janet M. Storey for her expertise with all live-animal work, and her tireless contributions to manuscript revision.

Additional Information and Declarations

Competing Interests

Author Contributions

Ethics

Kenneth B. Storey is an Academic Editor for PeerJ.

Christopher A. Dieni performed the experiments, analyzed the data, wrote the paper, prepared figures and/or tables.

Kenneth B. Storey conceived and designed the experiments, contributed reagents/materials/analysis tools, reviewed drafts of the paper.

The following information was supplied relating to ethical approvals (i.e., approving body and any reference numbers):

Conditions for animal care, experimentation, and euthanasia were approved by the Carleton University Animal Care Committee (B09-22) in accordance with guidelines set down by the Canadian Council on Animal Care.

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
