# Peer review of "Protein kinase C in the wood frog, Rana sylvatica: reassessing the tissue-specific regulation of PKC isozymes during freezing"

_PeerJ, doi:10.7717/peerj.558_

## Round 0.1 · original submission · Major Revisions

Please address all of the concerns of both reviewers. In particular, the concerns of Reviewer 2 are significant.

·

Basic reporting

I thought this was a generally well written and competent piece of work that conforms with the review criteria of the journal. I have just a few comments.

Experimental design

OK

Validity of the findings

OK

Additional comments

I found the use of hyphens, rather than commas, to denote a subordinate clause confusing in places. I suggest the use of commas for this purpose.

l. 261 something is wrong with this sentence

l. 366 conclusively

I wonder if there is a better way of quantifying the intensity of the immunoblots in situations where the band is undetectable in either frozen or the control sample. In Fig. 2 for example the change between control and frozen for both phospho-PKCδ (Thr505) and phospho-PKCθ (Thr538) are given as “--“ in Table 1 but the change is obviously greater in the latter. Perhaps the absolute change in intensity could be given, rather than the percentage, since where the band is undetectable intensity=0.

Reviewer 2 ·

Basic reporting

In general, the article is well-written. However, there are major concerns with regards to the presentation and depth of the background material provided on PKC-dependent signal transduction. At no point has the author updated the model of PKC maturation and signaling using more recent topical reviews. The limitations of using phosphorylation as a proxy for PKC signalling has not been fully addressed and the correlations between previous studies and PKC signalling need to be expanded upon.

Minor editorial comments:
Line 197: Do the authors mean PDK1 (phosphoinositide-dependent protein kinase 1) or PKD1 (protein kinase D 1)?
Line 289: PCKβII should be changed to PKCβII

Experimental design

Protein Kinase C (PKC) maturation (not activation) is an extremely complex and isozyme-specific cellular process that cannot be probed by monitoring single-site phosphorylation states. The lack of correlation in biochemical and phospho-PKC assays has been critiqued extensively, including in the recent review by Wu-Zhang and Newton (2013; Biochem J, 452, 195-209): “[c]anonically, PKCs are activated not by phosphorylation at these sites [referring to the activation loop, turn motif, and hydrophobic motif], which occurs constitutively, but by their acute translocation to membranes via second messenger-mediated membrane binding by their regulatory domains, an event that removes the pseudosubstrate [autoinhibitory sequence] from the active site”. Consequently, while the hypothesis that PKC signalling might be altered by freezing remains valid, the experimental design employed in this study is fundamentally flawed both conceptually and in execution.

Validity of the findings

There are major concerns with the interpretation of the data and conclusions made by the authors. This is due in large part to the technical limitations of the experimental design. Most concerning of all, is the lack of antibody validation done in this study. Studying signal transduction in a non-mammalian organism can be difficult; however, the use of proper controls and stringent antibody validation is especially important when trying to look at signalling by specific catalytic isozymes.

Major critiques:

(a) All of the primary antibodies used have been raised for use in mammalian systems. Consequently, it would be expected that blocking peptides (ideally) and/or (at the least) positive controls using a mammalian cell-line or tissues extracts (ideally, run on the same gel) would be employed. In general, the selective presentation of single bands of an unknown size does not provide adequate details and ideally, the un-cropped film should be available for viewing as supplemental information.

(b) Instead of using antibodies that recognize phosphorylated ser/thr sites that are known to be dynamically regulated within cells, the tissue-specific expression patterns of the PKC isozymes important for this study should be assed using antibodies that detect the total endogenous amounts of each PKC isozyme.

(c) Additionally, the phosphorylation-specific antibodies used monitor different aspects of PKC maturation. To truly compare the PKC “state” within the cell, antibodies against one (or, ideally, all) of the activation loop, turn motif, or hydrophobic motif should be used. How do you interpret and compare changes in the phosphorylation of the activation loop of one PKC isozyme with changes in the phosphorylation of the hydrophobic motif of another?

(d) Are all of the 15 PKC isozymes alluded to in the introduction present in amphibians at the genomic level? Perhaps epitope-mapping could be used to suggest that the sequences of these PKC isozymes are conserved and that they are present at the transcript level in closely related models.

(e) In standard phosphorylation studies, the amount of phosphorylated target protein (PKC) should to be normalized to the amount of total target protein (PKC; both phosphorylated and un-phosphorylated) within each treatment; thereby removing the need for other loading controls. In fact, a perfect example of why using the coomassie stain as a loading control should not be employed in phosphorylation studies can be seen by comparing the summary of phospho-PKD and total PKD levels provided in Table 1. Normalizing the phosphorylated PKD to total PKD would drastically alter the interpretation of the results. Also, from a methods standpoint, being able to normalize the amount of modified target protein to the total amount of the target protein is preferred to normalizing to non-specific protein bands.

(f) Phosphorylation should be quantified using densitometry and presented in a bar graph for each tissue tested. The summary table is useful, but should not be presented in the place of the quantified data.

Additional comments

The priming phosphorylations in the activation loop, turn motif, and hydrophobic motif of PKC are thought to control PKC maturation and stability as opposed to directly control catalytic activity. The assertion that upstream changes in PDK1 phosphorylation and activation control PKC activity is also somewhat incorrect since PDK1, like PKC, is constitutively active (and for that matter, PDK1 is constitutively phosphorylated at Ser241). In fact, cells that lack PDK1 (experimental knockdown) do not show decreased PKC activation, but rather produce PKCs that are proteolytically-labile; suggesting that the priming phosphorylation in the activation loop prevents PKC isozyme degradation. As a result, only by monitoring both the phosphorylated and total endogenous PKC levels could you conclude whether or not PKC stability is altered by freezing.

It should also be mentioned that, in general, the single-site phosphorylation studies cannot be easily interpreted due to PKC isozyme-specific differences in the potential sequence and importance of the priming phosphorylations. Since the authors propose to focus on the PDK1-dependent signalling network, perhaps they should focus their studies on the activation loop as this is the only PKC site shown to be a direct target of PDK1. Otherwise, it is suggested that a more extensive review of the current literature is performed in order to reconcile this complexity.

Additional comments:
(a) IP3 is not a proxy for PKC sigalling as DAG, not IP3, activates a subset of PKC isozymes. Also, DAG has many known intracellular targets other than PKC, and can be rapidly converted into other intracellular lipids.
(b) Authors need to expand on the importance of the phosphorylation sites monitored for each of the PKC isozymes investigated. Also, information about the controversy concerning the correlation between cellular regulation of these sites and catalytic activity needs to be included. The results are only relevant if they can be framed in the context of the most current information on PKC signalling.
(c) What about the recent evidence that PKCδ does not require activation-loop phosphorylation for catalytic activity?
(d) What about the recent evidence for the importance of PKD phosphorylation at Ser916 and Ser744 for PKD signal termination, as opposed to PKD activation?

---

## Round 0.2 · Major Revisions

In the light of the two reviews of the revised manuscript: in order to publish this work, these experiments must be repeated following all the basic rules mentioned by the reviewers, or you must present the raw data for the experiments in the manuscript that demonstrates that all of the rules were followed by presenting raw images with all the samples in the same blots for comparison.

Reviewer 3 ·

Basic reporting

The article is clearly written and includes sufficient introduction and background to demonstrate how the work fits into the broader field of knowledge. The relevant literature is cited appropriately.

Experimental design

This paper uses western blot (WB) analysis to compare the levels of phosphorylated PKC isozymes in the tissues of normal versus frozen frogs. The conclusions reached by the authors depend entirely on a comparison of the intensity of bands from these WBs. However, the authors have not followed certain basic rules that are needed in order to ensure that the quantitative information derived from the WB analysis is reliable. Some of these are detailed below.

1. In all of the figures the bands stated to represent the phosphorylated PKC isozymes from the control and frozen frogs have been narrowly cut out of a WB and pasted individually into the figure. This selective data presentation raises a number of questions. First, it is not clear that the bands come from the same WB or have been selected from different WBs. In order to make a valid comparison between the levels of phosphorylated PKC in the control and frozen frogs, the samples should be part of the same WB so that it is clear that they have been processed and exposed in exactly the same way. Instead of individual cut-out bands, a direct comparison of band intensity requires that the authors show intact WBs containing both control and frozen samples.

2. Another problem with cutting out a very narrow section of the WB is that all information on the size of the band is lost. The authors state that “immunoreactive bands were confirmed to be PKC isozymes by comparing their approximate molecular weights to those listed on the manufacturer’s datasheet”. A section of the film large enough to contain the ladder should be shown, so that it is apparent that the bands are indeed of the correct size to be stated PKC isozyme. A larger section of the WB would also provide information on the specificity of the anti-phospho antibodies – whether they detected a single band or multiple bands.

3. Internal loading controls are not provided for any of the WBs. For publication quality work, loading controls are essential in order to normalize the data and compensate for differences in gel loading and transfer rates between samples. It is not possible to accurately compare the intensity levels of the bands within or between WBs without a loading control.

4. It is important that WBs be carried out using antibodies that detect the total unphosphorylated PKC isozyme. This would help to confirm the identity of the band detected by the anti-phospho antibody – that the PKC isozyme of interest is expressed in the tissue and is the correct size. WBs with total anti-PKC antibodies are also needed to show whether changes in the phosphorylation level simply reflect changes in the total amount of PKC present in the tissue.

Validity of the findings

The manner in which the data is presented, and the lack of appropriate controls, raise serious concerns whether the conclusions reached are valid.

Reviewer 4 ·

Basic reporting

The article by Dieni and Storey entitled “Protein kinase C in the wood frog, Rana sylvatica: reassessing the tissue-specific regulation of PKC isozymes during freezing” builds upon previous work by this group that suggested a role for PKC in freezing based on patterns of inositol 1,4,5 trisphosphate. The focus of this work is to explore the regulation of the PKC family by assessing the levels of phosphorylated forms of the PKC isozymes in various tissues of control and frozen frogs. The article is very well written, however, the data are uninterpretable due to flawed experimental procedure and a lack of controls.

The figure legends (Figs 1-4) do not define the experimental condition represented by the closed vs. open bars.

Experimental design

The western blot analysis, which is the basis of the entire paper, was not conducted appropriately. Samples must be run on the same western blot in order to make comparisons. This does not appear to be the case as single, separate bands are shown for each sample. There is a general lack of appropriate controls and thus inappropriate interpretation of the results (see below).

Validity of the findings

The manuscript is predicated on the findings of the western blot analysis from tissue samples extracted from control and frozen frogs. In order to make comparisons in expression levels between control (active) vs. frozen frogs, the samples must be run on the same western blot with molecular weight markers indicated. This is not the case for any of the experiments presented. The semi-quantitative densitometry data for each figure are problematic. The authors state that bands from coomassie stained immunoblots were used to normalize for total protein. This is not a conventional way to normalize for total protein and for many reasons in my opinion is not acceptable. A loading control such as actin or tubulin should be used. Importantly, the levels of total PKC isozymes (PKC α, β, θ, δ, ζ, λ) need to be assessed in order to determine if the changes in phosphorylated states of the isozymes are due to actual changes in phosphorylation rather than changes in the total levels of the PKC isozymes. For each tissue analyzed, a positive marker for that tissue should be included.

---

## Round 0.3 · accepted · Accept

Thank you for your hard work. I believe that you have responded appropriately to the comments of the reviewers.